# Milk Urea Concentration in Dairy Sheep: Accounting for Dietary Energy Concentration

**DOI:** 10.3390/ani9121118

**Published:** 2019-12-11

**Authors:** Valeria Giovanetti, Filippo Boe, Mauro Decandia, Giovanni Cristoforo Bomboi, Alberto Stanislao Atzori, Antonello Cannas, Giovanni Molle

**Affiliations:** 1Servizio Ricerca per la Zootecnia, AGRIS Sardegna, 07040 Olmedo, Italy; vgiovanetti@agrisricerca.it (V.G.); mdecandia@agrisricerca.it (M.D.); gmolle@agrisricerca.it (G.M.); 2Dipartimento di Agraria, Sezione di Scienze Zootecniche, Università di Sassari, Viale Italia 39, 07100 Sassari, Italy; filippoboe@gmail.com (F.B.); asatzori@uniss.it (A.S.A.); 3Dipartimento di Medicina Veterinaria, Università di Sassari, Via Vienna 2, 07100 Sassari, Italy; gcbomboi@uniss.it

**Keywords:** milk composition, dietary energy, dietary protein, N efficiency, nutrition

## Abstract

**Simple Summary:**

In this paper, we show that milk urea concentration (MUC) of dairy ewes is markedly affected not only by dietary protein concentration, as evidenced by previous research, but also by dietary energy concentration. Thus, to avoid misleading interpretations, the utilization of MUC as indicator of the protein status of ewes should account for the dietary energy concentration. Minimal, optimal, and maximal MUC values for different combinations of dietary energy and protein are proposed. Because frequent bulk tank MUC analysis is easy to perform and cost-effective, the reference values proposed here can be used for optimizing sheep milk and reproductive performances while curbing N release from excreta.

**Abstract:**

In dairy sheep milk urea concentration (MUC) is highly and positively correlated with dietary crude protein (CP) content and, to a lesser extent, with protein intake. However, the effect of dietary energy and carbohydrate sources on MUC of lactating ewes is not clear. Thus, the objective of this study was to assess the effects of diets differing in energy concentration and carbohydrate sources on MUC values in lactating dairy ewes. Two experiments were conducted (experiment 1, E1, and experiment 2, E2) on Sarda ewes in mid and late lactation kept in metabolic cages for 23 d. In both experiments, homogeneous groups of five ewes were submitted to four (in E1) or three (in E2) dietary treatments, consisting of pelleted diets ranging from low energy (high-fiber diets: 1.2–1.4 Mcal of net energy for lactation (NE_L_)) to high energy (high-starch diets: 1.7–1.9 Mcal of NE_L_) contents, but with a similar CP concentration (18.4% dry matter (DM), on average). Each diet had a different main ingredient as follows: corn flakes, barley meal, beet pulp, or corn cobs in E1 and corn meal, dehydrated alfalfa, or soybean hulls in E2. Regression analysis using treatment means from both experiments showed that the best predictor of MUC (mg/100 mL) was the dietary NE_L_ (Mcal/kg DM, MUC = 127.6 − 51.2 × NE_L_, R^2^ = 0.85, root of the mean squared error (rmse) = 4.36, *p* < 0.001) followed by the ratio CP/NE_L_ (g/Mcal, MUC = −14.9 + 0.5 × CP/ NE_L_, R^2^ = 0.83, rmse = 4.63, *p* < 0.001). A meta-regression of an extended database on stall-fed dairy ewes, including the E1 and E2 experimental data (n = 44), confirmed the predictive value of the CP/ NE_L_ ratio, which resulted as the best single predictor of MUC (MUC = −13.7 + 0.5 × CP/NE_L_, R^2^ = 0.93, rmse = 3.30, *p* < 0.001), followed by dietary CP concentration (MUC = −20.7 + 3.7 × CP, R^2^ = 0.82, rmse = 4.89, *p* < 0.001). This research highlights that dietary energy content plays a pivotal role in modulating the relationship between MUC and dietary CP concentration in dairy sheep.

## 1. Introduction

Blood urea concentration (BUC) and milk urea concentration (MUC) are currently used as nutritional indicators in ruminants, because they are closely related to digestive tract activity [1] and endogenous ammonia production [2], the latter being associated with gluconeogenesis. Because urea is the major end product of N metabolism in ruminants, blood and milk urea contents are good predictors of nitrogen excretions [3]. Blood urea concentration cannot be measured routinely, because sampling requires invasive techniques and its concentration can change rapidly after meals. On the contrary, MUC is more stable and easier to sample than BUC. In dairy cows, several studies have shown that MUC is related to dietary crude protein intake (CPI), percentage of rumen degradable and undegradable protein, and the protein-to-energy ratio in diet [4,5]. In dairy sheep fed diets ranging from 14% to 21% of dietary CP (dry matter (DM) basis), MUC was positively and linearly related to dietary CP content and, to a lesser extent, to protein intake [6]. In that experiment, a relatively narrow range of energy concentration was also tested (1.55–1.65 Mcal of net energy for lactation (NE_L_)), and energy was not found correlated with MUC. This contrasts with previous findings on dairy cattle [7] and goats [8]. However, in a later study [9] comparing diets with 1.40 and 1.59 Mcal/kg DM of NE_L_ and 19–20% DM of CP fed to mid-lactation ewes, a significantly lower MUC in the ewes fed the diet with higher energy content was found.

There are no studies directly testing the effect of dietary energy concentration on MUC in dairy sheep. Because MUC reference values in lactating ewes are substantially higher compared with those in lactating goats and cows [10], it is worth exploring the effects of factors other than crude protein content and intake on MUC in lactating ewes, with particular reference to dietary energy level and source (fiber vs. non-fiber carbohydrates). 

Prediction equations obtained from individual experiments can be easily compared with findings from other studies by using a meta-regression or meta-analysis approach [11]. It can be useful to check the robustness of the observed relationships and gather a more generic algorithm to be empirically used in broad contexts and at the field level or for other modeling purposes [12]. Comprehensive analysis of experimental data identified the dietary CP concentration as the main factor influencing MUC in dairy cows [13,14], whereas no similar efforts have been carried out for sheep.

Thus, this study was carried out with two main objectives: (i) assessing the relationships between MUC and the dietary content or intake of nutrients in dairy ewes fed diets characterized by a wide range of energy contents and carbohydrates sources; and (ii) comparing the same relationships within a meta-regression based on a broader database inclusive of other studies on stall-fed dairy sheep.

## 2. Materials and Methods 

### 2.1. Animals and Diets

The study was conducted at the Bonassai experimental farm of the Agricultural Research Agency of Sardinia (AGRIS Sardegna), located in the northwest of Sardinia (40° N, 32° E, 32 m a.s.l.), in Italy. The animal protocol described below was fully in compliance with the European Union (EU) and Italian regulations on animal welfare and experimentation, and it was approved by the veterinarians responsible of the ethic and welfare control in animal experimentation of AGRIS and the University of Sassari. All measurements were taken by personnel previously trained and authorized by the institutional authorities on ethical issues both from AGRIS and the University of Sassari.

The study consisted of two feeding experiments conducted on Sarda dairy sheep during mid (in March) and late (June–July) lactation. In experiment 1 (E1), four complete pelleted diets were tested on 20 ewes, and in experiment 2 (E2), other three complete pelleted diets were tested on 15 ewes. Each experiment consisted of a seven-day preliminary period, a fourteen-day adaptation period, and a nine-day experimental period. During the preliminary period, the ewes of each experiment grazed ryegrass-based pastures. were supplemented with a mixture made of equal proportions of the relative experimental pelleted diets for four days, and were then confined in pens for three days, during which time they received only the mixture of the experimental pelleted diets for three days. After the preliminary period, the ewes were allocated to homogeneous groups and put in individual metabolic cages for the adaptation and experimental periods. 

In E1, the 20 mid-lactation ewes were allocated to four homogeneous groups of five animals each on the basis of their days in milk (DIM; Table 1; mean ± s.d.), milk yield (MY), body weight (BW), body condition score (BCS), age, and parity. The ewes were fed the pelleted diets ad libitum in two daily meals. The same animals were then re-randomized and used in late lactation. In E2, 15 mid-lactation ewes were assigned to three homogeneous groups of five animals each, on the basis of their DIM (Table 1), MY, BW, BCS, age, and parity. The same animals were then re-randomized and used in late lactation.

Throughout the period between the mid- and late-lactation measurements, the ewes were fed at pasture and machine-milked twice a day at 07:00 h and 15:00 h in a milking parlor. The animals were machine-milked twice a day at 07:00 h and 15:00 h inside the cages during the adaptation and experimental periods. All the animals had ad libitum access to water throughout the study.

The ingredients and the chemical composition of the diets used in E1 and E2 are summarized in Table 2. On the basis of their main ingredient, the following diets were tested: CF = corn flakes, BM = barley meal, BP = beet pulp, and CC = corn cobs, in E1; and CM = corn meal, AA = dehydrated alfalfa, and SH = soybean hulls, in E2. All the diets contained dehydrated alfalfa as a common base and other ingredients (barley meal, corn flakes, corn meal, beet pulp, corn cobs, corn germ, corn gluten meal, soybean hulls, wheat middlings, minerals, and vitamins) were added in order to obtain different fiber (neutral detergent fiber (NDF), acid detergent fiber (ADF), and acid detergent lignin (ADL)) and energy contents, while keeping CP concentrations similar.

The energy content of the diets was calculated as net energy for lactation (NE_L_) on the basis of total digestible nutrient (TDN, % DM) [15]:NE_L_ (Mcal/kg DM) = 0.0245 TDN − 0.12,(1)
whereas TDN was calculated as follows:TDN (% DM) = (dCPI + dNDFI + dNFCI + dEEI × 2.25)/DMI × 100(2)
where dCPI = digestible CP intake (g/day), dNDFI = digestible NDF intake (g/day), dNFCI = digestible non-fiber carbohydrates (NFC) intake (g/day), dEEI = digestible ether extract intake (g/day), and DMI = dry matter intake (g/day). The data on intake of digestible nutrients measured in vivo and used in the above equations are reported in [16]. In order to prevent acidosis in late lactation, due to a possible uneven feeding pattern associated with the high diurnal temperature typical of the late-lactation period (June–July), 10 g/day per head of sodium bicarbonate were added to all the diets.

### 2.2. Measurements

In both experiments, the individual intake was measured by weighing the offered diets and the corresponding orts 24 h after the first daily meal during the experimental period. Samples of feed on offer were collected once a week and stored until analyses. Individual milk yield was measured three times during the experimental period, and individual milk samples, at the morning and afternoon milking, were also taken. The concentration of protein digestible in the intestine, when energy or nitrogen is not limiting rumen microbial growth (PDIN and PDIE), was calculated for each diet using tabular values [17]. The requirements of dairy sheep in terms of protein digestible in the intestine (PDI) were estimated with the equation of [18], and PDIN balance was then estimated as the difference between intake and requirements (g/day) and as the ratio between PDI balance (g/day) and PDI requirements (g/day), expressed as a percentage.

### 2.3. Chemical Analyses

Feed samples were analyzed for DM; ash; CP; ether extract (EE, [19]); ash-free fiber fractions as NDF, without the use of sodium sulfite, ADF, and ADL [20]; CP fractions [21]; and starch (polarimetric method, [22]). The non-fiber carbohydrates (NFC) concentration was calculated as [100 − (NDF − NDIP) − CP − EE − ash], where NDIP = neutral detergent insoluble protein. Individual milk samples were analyzed for fat, true protein, and lactose using the infrared method (Milkoscan 4000, Foss Electric, Hillerød, Denmark) and urea by a colorimetric method (ChemSpec 150, Bentley Instruments Inc., Chaska, MN, USA; Broutin, 2000) calibrated by differential pH measurements (CL10-Eurochem, Rome, Italy).

### 2.4. Statistical Analysis

To target the first objective of the study (assessment of the effect of dietary energy on milk urea concentration), the results of the experiments were averaged by dietary group and physiological stage, and treatment means (n = 14) of the two experiments were pooled and used to study the relationships between dietary variables, nutrient intake, and MUC, as detailed below. First, values of MUC were regressed against dietary concentration and intake of nutrients, as well as PDIN balance, using a simple linear regression model:Yi = B_0_ + C_0_ X_i_ + e_i_(3)
where B_0_ = intercept, C_0_ = regression coefficient, Xi = independent variable, and ei = random error. Second, a stepwise regression analysis was performed to verify if any multiple regression model could fit better than simple regression models to predict MUC. All dietary variables already quoted were tested using *p* < 0.15 as entry and stay probability thresholds. Because no variables were kept in the model, except for the content of dietary energy, no further attempts were made to test multiple regression models.

To pursue the second objective of the study (meta-analysis of the available literature on stall-fed dairy sheep) an extended dataset was made by adding to the current experiment results data from other experiments where dietary CP and energy intake and or their contents in sheep diet were related to milk urea. The search of relevant papers was done using Scopus with the keywords “sheep and nutrition and milk urea”, exploring all the scientific literature available to those engines in the time range of 1970–2019. Overall, the search resulted in 25 papers in the Scopus database, including primary and secondary documents, most of which were not relevant due to the focus on non-nutritional aspects or to the lack of accurate information on CP and energy intake (mostly grazing studies) or because urea was measured erratically or only in sheep plasma. Furthermore, all experiments that included diets containing tannins were discarded from the dataset because tannins are known for of their effects in modulating the use of dietary proteins. Low to moderate levels of tannins in the diet may actually reduce the protein degradation in the rumen and increase amino acid flow to the small intestine, while high levels can reduce voluntary feed intake and nutrient digestibility. At the end of this screening process, only six studies on stall-fed dairy sheep reported in the literature were selected [6,9,23,24,25,26], and their treatment means merged to those obtained in our study to form the extended dataset.

The relationships derived from the extended dataset were calculated using two statistical models: (1) simple linear regression models and (2) meta-analytical mixed models [27], which included the regressors as fixed effects and the “study effect” (Exp) as random effect [11]. In particular, the implemented model was as follows:Y_ij_ = A_0_ + B_0_ X_ij_ + Exp_i_ + b_i_ X_ij_ + e_ij_(4)
where A_0_ = overall intercept, B_0_ = overall regression coefficient, X_ij_ = independent variable, Exp_i_ = random effect of the study on the intercept, b_i_ = random effect of the study on the regression slope and e_ij_ = random error.

## 3. Results

### 3.1. Composition of the Diets

The composition of the diets (Table 2) corresponded to the planned values, except for CP concentration, which unexpectedly showed a range of variability, although it was limited to about ±5% of the average within the trial (range of CP: 17.9–19.8% in E1 and 17.5–19.7% in E2, DM basis). In contrast, the range of NDF (28.5–45.8% in E1, 23.9–51.9% in E2, DM basis) and starch concentration (8–28.8% in E1, 3.2–35.6% in E2, DM basis) was very broad. Therefore, as planned, the diets were characterized by contrasting levels of dietary energy, spanning across the two experiments from a minimum of 1.2 Mcal of NE_L_/kg DM to a maximum of 1.9 Mcal of NE_L_/kg DM.

### 3.2. Animal Data

Mean values of the animal data used in the subsequent regression analysis and their ranges are displayed in Table 3.

### 3.3. Results of E1 and E2

Significant negative and linear relationships were found between dietary NFC or starch and MUC (R^2^ = 0.39, *p* < 0.01; Table 4). The strength of the relationship markedly increased when MUC was regressed against dietary NE_L_ (R^2^ = 0.85, *p* < 0.01), which was the best single predictor of MUC.

The dietary CP content tended to be related to MUC (*p* = 0.09). In contrast, the ratio CP/NE_L_ was strongly related to MUC, performing similarly to dietary NE_L_ (R^2^ = 0.83, *p* < 0.001). The sum of A + B1 protein fractions was more closely related to MUC (R^2^ = 0.53) than the single fractions B1 (R^2^ = 0.49) or A (R^2^ = 0.35). 

Significant relationships were found when MUC was regressed against the fiber fractions, among which ADL showed the highest coefficient of determination (R^2^ = 0.57), followed by ADF (R^2^ = 0.42) and NDF (R^2^ = 0.27). The relationships between NE_L_I or CPI and MUC were not significant (data not shown). NDFI was not related to MUC either. On the contrary, NFCI had a weak but significant negative relationship with MUC (R^2^ = 0.36, *p* < 0.01), which was also related negatively to starch intake (starchI, R^2^ = 0.48, *p* < 0.003). A linear positive relationship was found between PDIN balance, expressed as g/day, and MUC (R^2^ = 0.27, *p* < 0.056). The strength of the relationship markedly increased when MUC was regressed against PDIN balance, expressed as % of PDIN requirement (R^2^ = 0.54, *p* < 0.003).

### 3.4. Results of the Meta-Analyses

The extended database on stall-fed sheep had a total of 44 dietary treatments, characterized by a wide range of dietary CP and energy contents (CP: from 12.3% to 24.6% DM; NE_L_: from 1.20 to 1.88 Mcal/kg DM). The diets were based on pelleted concentrates, hay and concentrates, and fresh forages clipped at a height of 5 cm above the soil surface. The trend of MUC and the variation of dietary CP, NE_L_, or CP/NE_L_, considering each experiment separately, are depicted in Figure 1, Figure 2 and Figure 3, respectively.

The results of regression meta-analyses are shown in Table 5.

Both fixed and mixed regression models indicated a strong linear relation between dietary CP and MUC (R^2^ = 0.82 and 0.68 for mixed and fixed models, respectively, Table 5). The close relationship between NE_L_ and MUC found after the analysis of pooled data from E1 and E2 (Table 4) was confirmed by the meta-analysis, with similar intercepts and slopes (R^2^ = 0.73 and 0.46, for mixed and fixed models, respectively). The ratio of CP/NE_L_ was the best single predictor of MUC. The equations estimated by the two models were similar for intercept and slope, whereas R^2^ was slightly higher in the mixed than in the fixed model (0.93 vs. 0.88). The slopes of these equations were similar to those reported after E1 and E2 regression analysis (Table 4). A positive relationship was found between NDF and MUC, but only when the mixed model was implemented. As expected, when MUC was regressed against NFC, the slopes were negative and identical between the two models (*p* < 0.001, Table 5). As regards the intake of nutrients, only when MUC was regressed against CPI did the relationship became significant, with a higher determination coefficient and lower root of the mean squared error (rmse) in the mixed model (R^2^ = 0.76, *p* = 0.001, Table 5).

## 4. Discussion

In both experiments, the small particle size of the diets did not seem to impose physical constraints to DMI, which was probably regulated mostly by the energy demand. The overall lower DM and CP intake values observed in late-lactation ewes were probably due to the lower requirements of the animals with decreased milk yield, typical of this stage; particularly high MUC values were probably due to an excess of protein concentration compared with the needs of the animals.

### 4.1. Results of E1 and E2

Average group data were used in these analyses, because MUC urea is usually sampled for groups of ewes and not individually and for the need of developing relationships comparable with those developed in the meta-analysis, based on treatment means in the literature. 

The non-significant relationship between MUC and dietary CP concentration (Table 4) was very likely due to the small variation in dietary CP considering both experiments (from 17.5% to 19.8% of DM; Table 2). Milk urea concentration was more related to the soluble N fractions of the diets than to total dietary CP. In particular, the B1 protein fraction was more strongly associated with MUC than the A protein fraction, despite its lower presence in the diets. This result could be explained by (i) the wider range of variation of the fraction B1 in the diets under study and (ii) the likely variability in its utilization at the rumen level. Indeed, while fraction A is usually completely fermented in the rumen, part of fraction B1 can escape the rumen, depending on the combination of its degradation and passage rates [28], which in turn affects the partitioning of potentially degradable CP into rumen degradable (RDP) and rumen undegradable protein (RUP). Indeed, the dietary percentage of RDP and RUP influenced milk urea N in sheep fed almost isoenergetic diets, with higher milk urea N values in ewes fed 14% RDP and 4% RUP (DM basis) than in those fed 12% RDP and 4% RUP (DM basis, [25]).

The significant regression of MUC against dietary NE_L_ found in this study on dairy sheep (Table 4) is in agreement, as a general trend, with previous studies on dairy cows [7] and goats [8]. This result suggests that enhancing the energy content of the diet increases the uptake of N by rumen microorganisms and reduces amino acid gluconeogenetic utilization, thus reducing the wastage of N. In the present study, the range of dietary NE_L_ was set to be much wider than in other previously cited experiments conducted on sheep [6,9] in order to cover the range of the energy densities frequently experienced by lactating dairy sheep (from 1.20 to 1.95 of NE_L_ per kg DM, [29]). Interestingly, although MUC was highly associated with NE_L_, it was poorly associated with dietary NFC and NDF, suggesting that the overall energy availability was more important as a determinant of MUC than the carbohydrate sources (fibrous or starchy) from which the energy was derived.

In this study, the passage rate was high for all the diets due to the small particle size of the pellets, as reported elsewhere [16,30]. This suggests that fermentable energy in the rumen was probably the main limiting factor for N utilization, as also indicated by the negative relationship between NFC content and MUC and the positive relationship between fiber fractions concentrations and MUC found in our study. Another reason for high levels of MUC in the sheep fed high-fiber diets could be the poor synchronization between energy and N supply at the rumen level, as shown in sheep [31].

As expected, MUC was strongly and positively related to PDIN balance (expressed as %), in agreement with the results obtained in Saanen goats [8]. These authors reported a regression equation with a higher coefficient of determination (R^2^ = 0.92) than that found in our experiment (R^2^ = 0.54). Interestingly the slopes of the regressions for dairy sheep and goats were similar (0.28 and 0.34 mg/dL, respectively), unlike the intercepts (26.9 and 22.9 mg/dL, respectively). The lower value of the intercept found in the goat regression could be ascribed to their more efficient recycling of urea from blood to the rumen in goats than sheep [32]. According to this author, the greater secretion of saliva and the broader rumen surface for NH_3_ absorption in goats would explain the differences in MUC values among these animal species. Another possible explanation of the higher values of MUC in sheep compared with goats (and cattle) is their higher consumption of sulfur-containing amino acids for wool production. Therefore, sheep have a lower efficiency of conversion of metabolizable protein to net protein and thus higher ammonia wastage than cattle and goats, as suggested by [33] and [10]. 

### 4.2. Results of the Meta-Analyses

Pooling the data of E1 and E2 with those from other studies allowed to evaluate to what extent the relationships found in the present research fit a broader database (Table 5). The strict relationship (R^2^ = 0.82, *p* < 0.001) between dietary CP and MUC observed after meta-analyses confirms that the weak relationship between these variables found in E1 and E2 was strongly determined by the small range of CP variation between the experimental diets. Indeed, when the experiments considered a wide range of dietary CP concentration (studies 1, 6, and 7; Figure 1), the relationship between MUC and CP was tight and linear. This did not occur in the experiments with a small range of dietary CP concentration (studies 2, 3, 4, 5, and 8, Figure 1), where other experimental variables were likely more influential. Similarly, the linear relationship between MUC and NE_L_ was evident only when a wide range of NE_L_ was considered (experiments E1 and E2, which correspond to studies 2 and 3, respectively, in Figure 3). In contrast, all the studies included in the reviewed dataset were characterized by a good relationship between MUC and CP/NE_L_ ratio (Figure 2). Moreover, the residual plot distribution from the mixed model analysis, which includes the random effect of study, showed that the distribution of residual errors was closer to zero when CP/NE_L_, rather than CP or NE_L_, was used as the single predictor of MUC. The adjusted values of MUC based on the model residuals and the equation obtained from the meta-regression model plotted against CP/NE_L_ values are shown in Figure 4.

This is in agreement, as a general trend, with what it was found in a study on dairy cows [4], suggesting that the ratio between dietary protein and energy content is more related to MUC than their singular concentrations or intakes.

The above considerations confirm that MUC can have a practical application for assessing the adequacy of protein nutrition in dairy ewes [10] and also point out that the ‘modulation effect’ of dietary energy has to be taken into account. According to [34], other dietary components, such as tannic phenols, can contribute to this modulation, particularly when ruminants are exposed to plant secondary metabolites, which is a common situation under grazing conditions.

### 4.3. Practical Applications

Milk urea concentration in dairy sheep farms can be easily and cost-effectively analyzed in bulk tank samples. This suggests that MUC can be used as a diagnostic tool to monitor the nutritional status of groups of lactating ewes in view of optimizing both milk [6] and reproductive performance [35], while curbing the release of N from excreta.

Based on the results of the meta-analysis (Table 5), MUC can be estimated according to the following equation: MUC (mg/100 mL) = −13.7 + 0.5 CP/NE_L_ (g/Mcal).(5)

This equation was used in Table 6 to predict the level of MUC corresponding to different CP and NE_L_ concentrations in the diet of lactating ewes. Thus, as long either CP or and NE_L_ concentration is known, the other variable can be predicted by measuring MUC. 

The grey area reported in Table 6 represents a risk zone; it reports MUC values close to or higher than 56 mg/100 mL, indicated by [35] as a threshold above which there is high probability of impairment of reproductive function in sheep with a decrease of conception rate.

The MUC levels, reported in Table 6, are higher than the corresponding estimates based on the regression equation developed by [6] using their experimental data and data from the literature, except for diets with high NE_L_ concentration (equal to or higher than 1.6 Mcal/kg of DM) and CP concentration (equal to or higher than 15% CP, DM basis), for which the values are similar between the two studies. 

This discrepancy may be related to the fact that the database used by Cannas et al., 1998 [6] to develop their equations included not only milk urea data but also plasma urea data from experiments on non-lactating sheep, which had much lower dietary CP and CPI than the lactating animals. In contrast, our extended database is focused on milk urea, by itself usually higher than blood urea at equal nutritional conditions, from lactating ewes fed, in most cases, diets characterized by a positive PDI balance and high dietary CP and CPI. Indeed, not only the dietary CP but also the CPI positively affects MUC (Table 5).

In addition, [6] considered both blood and milk urea in their relationship, thus including in their analysis dietary treatments applied to dry ewes characterized by lower protein levels than those typical of lactating sheep rations. It is noteworthy that MUC levels as high as 50 mg/dL are not uncommon in sheep flocks grazing immature pasture with a high content of readily fermentable N. Therefore, the incorporation of high CP dietary treatments in the meta-analysis seems an important way to account for these excess conditions, frequently encountered in commercial dairy flocks.

If the independent and the dependent variables of the equation above are swapped around, the following equation is obtained:CP/NE_L_ (g/Mcal) = 39.96 + 1.61 MUC (mg/100 mL); rmse = 5.78, R^2^ = 0.93, *p* < 0.001.(6)

Equation (6) indicates that measurements of MUC can provide an accurate prediction of the CP/NE_L_ ratio of the diet. Thus, if either CP or NE_L_ concentrations are known, MUC can be used to estimate the other unknown variable, e.g., by interpolation of data in Table 6.

## 5. Conclusions

The experiments undertaken in this study found a marked negative linear relationship between dietary energy content and milk urea level in dairy ewes in mid and late lactation. The dietary NE_L_ content was the best singular predictor of MUC, closely followed by the CP/NE_L_ ratio. In contrast, MUC was not associated with dietary CP content because of the very small range of CP variation between the tested diets. The relevance of the dietary energy content and of the CP/NE_L_ ratio, as predictors of MUC, was confirmed after the meta-analyses of the extended database.

Further research is needed to improve the value of MUC as a nutritional index in the management of feeding in dairy sheep, especially under grazing conditions not considered in this research.

## Figures and Tables

**Figure 1 animals-09-01118-f001:**
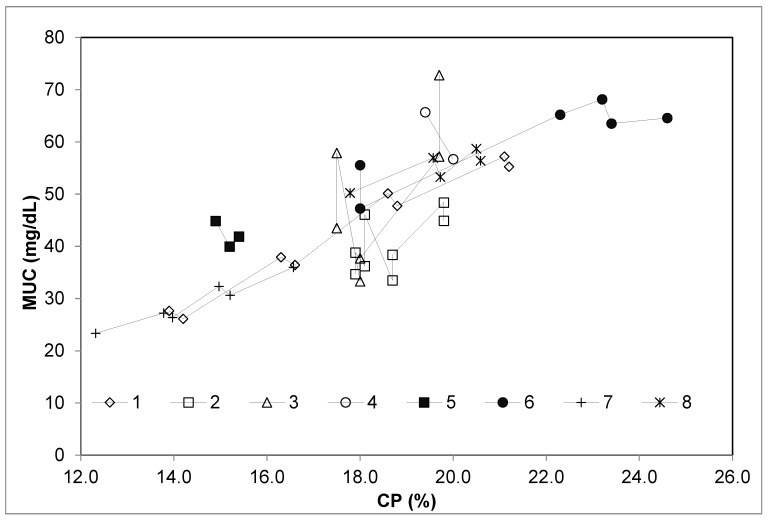
Relationship between milk urea concentration (MUC; mg/dL) and crude protein (CP; % DM) considering each single experiment on dairy sheep of the meta-analysis database separately. 1 = [6]; 2 = E1, present study; 3 = E2, present study; 4 = [9]; 5 = [24]; 6 = [23]; 7 = [26]; and 8 = [25].

**Figure 2 animals-09-01118-f002:**
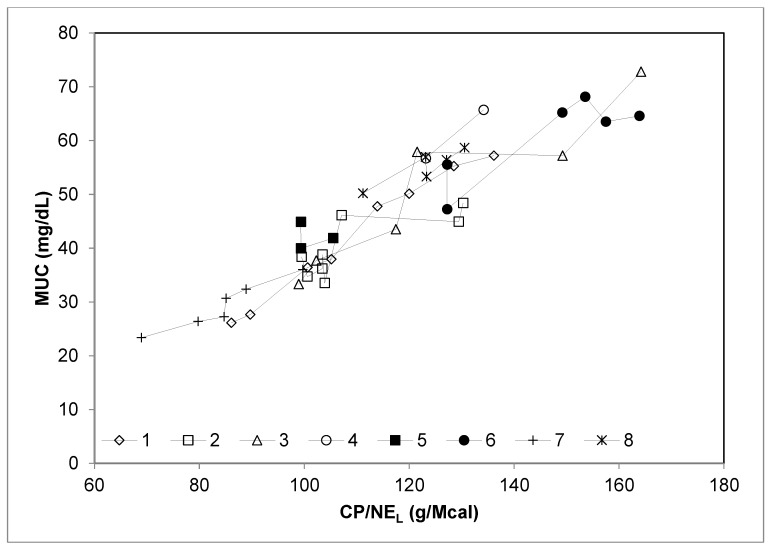
Relationship between milk urea concentration (MUC; mg/dL) and the ratio among crude protein (CP) and net energy for lactation (CP/NE_L_, g/Mcal) considering each single experiment on dairy sheep of the meta-analysis database separately. 1 = [6]; 2 = E1, present study; 3 = E2, present study; 4 = [9]; 5 = [24]; 6 = [23]; 7 = [26]; and 8 = [25].

**Figure 3 animals-09-01118-f003:**
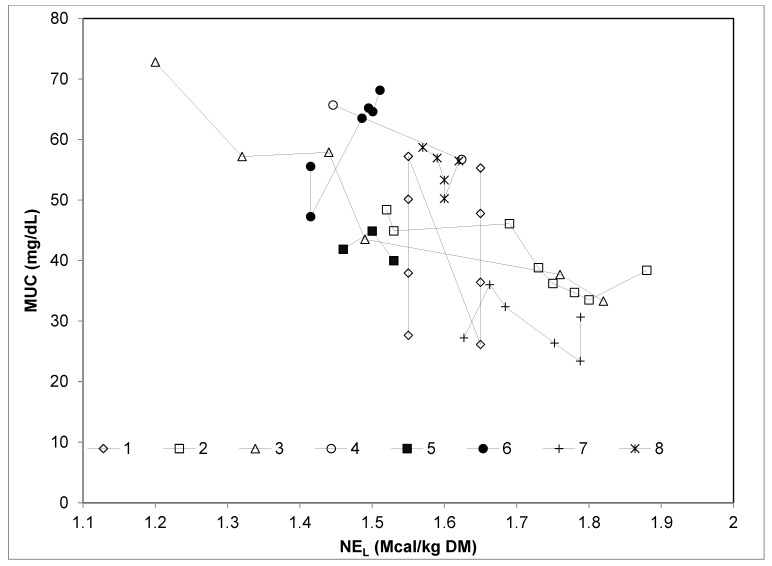
Relationship between milk urea concentration (MUC, mg/dL) and net energy for lactation (NE_L_, Mcal/kg DM) considering each single experiment on dairy sheep of the meta-analysis database separately. 1 = [6]; 2 = E1, present study; 3 = E2, present study; 4 = [9]; 5 = [24]; 6 = [23]; 7 = [26]; and 8 = [25].

**Figure 4 animals-09-01118-f004:**
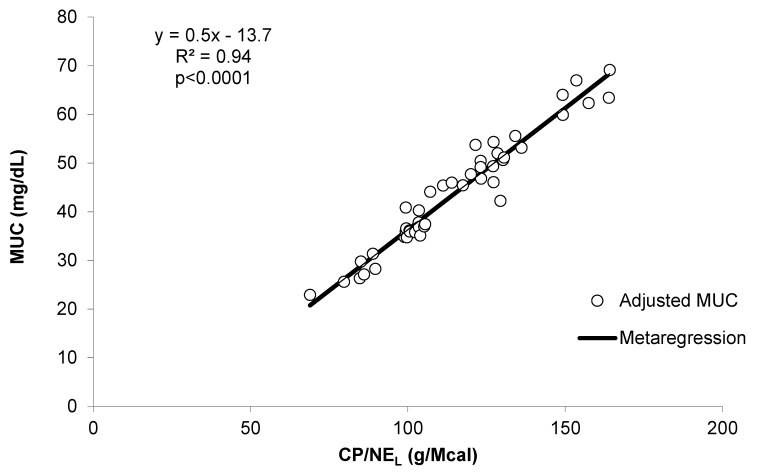
Relationship between milk urea concentration (mg/dL) and the ratio among crude protein and net energy for lactation (CP/NE_L_, g/Mcal) as obtained from the meta-regression analysis.

**Table 1 animals-09-01118-t001:** Description of animals used in the experiments E1 and E2.

Item	Experiment 1		Experiment 2	
	Mid Lactation	Late Lactation	Mid Lactation	Late Lactation
DIM ^1^ (days)	118 ± 3	222 ± 3	119 ± 11	224 ± 10
MY ^2^ (mL/day)	1792 ± 259	1516 ± 311	1401 ± 207	1011 ± 349
BW ^3^ (kg)	43.1 ± 3.5	47.6 ± 4.3	44.6 ± 4.7	46.2 ± 3.4
BCS ^4^ (0–5)	2.3 ± 0.18	2.6 ± 0.15	2.4 ± 0.25	2.5 ± 0.17
Age (years)	4 ± 1	4 ± 1	5 ± 2	5 ± 2
Parity (n.)	3 ± 1	3 ± 1	4 ± 2	4 ± 2

^1^ DIM = days in milk.; ^2^ MY = milk yield.; ^3^ BW = body weight.; ^4^ BCS = body condition score.

**Table 2 animals-09-01118-t002:** Ingredients and chemical composition of the experimental diets.

Item	Experiment 1				Experiment 2		
	CF ^1^	BM	BP	CC	CM	AA	SH
Ingredients							
Barley meal (% DM)		47.0					
Corn flakes (% DM)	51.6						
Corn meal (% DM)					59.3		
Dehydrated alfalfa (% DM)	27.0	28.0	26.0	26.6	26.7	89.4	26.2
Beet pulp (% DM)			40.9				
Corn cobs (% DM)				32.5			
Corn germ (% DM)	6.4	12.9	20.0	20.0			
Corn gluten meal (% DM)	8.0	1.7	1.9	5.8	8.2		5.5
Soybean hulls (% DM)							62.3
Wheat middlings (% DM)			6.1	10.0		5.0	
Minerals and vitamins (% DM)	7.0	7.0	5.1	5.1	5.8	5.6	6.0
Chemical composition ^2^							
CP (% DM)	18.7	18.1	17.9	19.8	18.0	19.7	17.5
EE (% DM)	2.7	3.1	3.3	4.4	3.0	2.6	2.5
Ash (% DM)	8.1	9.3	8.3	8.5	7.3	11.2	9.0
NDF (% DM)	28.5	33.2	44.9	45.8	23.9	45.6	51.9
ADF (% DM)	14.2	16.3	21.4	22.2	12.4	29.2	36.0
ADL (% DM)	2.8	3.0	3.3	3.7	2.1	6.0	2.8
NFC (% DM) ^3^	47.3	41.0	31.8	26.1	51.2	25.1	23.6
WSC (% DM)	3.7	3.7	4.5	3.6	3.3	4.3	3.2
starch (% DM)	28.8	25.2	8.0	12.4	35.6	7.1	3.2
Ca (% DM)	1.18	1.42	1.25	0.88	0.78	1.62	1.17
P (% DM)	0.55	0.45	0.50	0.52	0.49	0.51	0.55
Cl (% DM)	0.62	0.64	0.60	0.64	0.62	0.96	0.63
K (% DM)	0.98	1.22	0.90	1.14	0.97	2.43	1.63
Mg (% DM)	0.25	0.30	0.33	0.31	0.22	0.30	0.27
S (% DM)	0.60	0.53	0.57	0.58	0.64	0.58	0.56
Na (% DM)	0.62	0.74	0.72	0.58	0.60	0.62	0.60
NE_L_ mid lactation (Mcal/kg DM)	1.80	1.75	1.78	1.52	1.82	1.32	1.49
NE_L_ late lactation (Mcal/kg DM)	1.88	1.69	1.73	1.53	1.76	1.20	1.44
CP/NE_L_ mid lactation	104	103	101	130	99	149	118
CP/NE_L_ late lactation	99	107	103	129	103	165	123
Protein fractions ^4^ (% DM)							
A	2.9	3.5	3.5	4.4	3.3	4.7	3.0
B_1_	1.3	1.7	1.3	1.3	0.7	2.4	1.4
B_2_	9.1	8.2	6.8	9.5	10.6	8.4	8.6
B_3_	3.4	3.2	4.4	2.9	1.9	2.7	2.6
C	1.9	1.5	1.8	1.7	1.5	1.5	1.9
PDIN ^5^ (% DM)	132	104	100	112	137	114	120
PDIE ^6^ (% DM)	137	106	107	109	147	94	129

^1^ Diets (based on main ingredients): CF = corn flakes; BM = barley meal; BP = beet pulp; CC = corn cobs; CM = corn meal; AA = dehydrated alfalfa; SH = soybean hulls.; ^2^ DM = dry matter; CP = crude protein; EE = ether extract; NDF = neutral detergent fiber; ADF = acid detergent fiber; ADL = acid detergent lignin; NFC = non-fiber carbohydrates; WSC = water soluble carbohydrates; NE_L_ = net energy for lactation.; ^3^ NFC = 100 − (NDF − NDIP) − CP − EE − Ash, where NDIP = neutral detergent insoluble protein (% DM).; ^4^ protein fractions: A = non-protein nitrogen (NPN); B_1_ = buffer soluble true protein; B_2_ = buffer insoluble protein − neutral detergent insoluble protein; B_3_ = neutral detergent insoluble protein − acid detergent insoluble protein; C = acid detergent insoluble protein.; ^5^ PDIN balance (g/day) = PDIN allowance (g/day) − PDIN requirement (g/day).; ^6^ PDIN balance (%) = (PDIN allowance (g/day) − PDIN requirement (g/day))/PDIN requirement (g/day) × 100.

**Table 3 animals-09-01118-t003:** Intake of dry matter and nutrient, PDIN balance, milk yield and composition, and milk urea concentration (mean and range) in mid- and late-lactation dairy ewes in experiments 1 and 2.

Item ^2^		Experiment 1								Experiment 2					
		Mid Lactation				Late Lactation				Mid Lactation			Late Lactation		
		CF ^1^	BM	BP	CC	CF	BM	BP	CC	CM	AA	SH	CM	AA	SH
DMI (g/day)	Mean	1870	1849	2055	2602	1295	1752	1992	2182	1717	2196	2634	890	1903	1603
	Min	1262	1674	1585	2159	1192	1024	1701	1863	1413	1933	2136	582	1701	1230
	Max	2401	2238	2588	3392	1367	2423	2386	2548	2096	2438	3175	1211	2148	1879
CPI (g/day)	Mean	349	334	367	516	241	317	356	433	310	432	461	160	375	280
	Min	235	302	283	428	222	185	304	369	255	381	374	105	335	215
	Max	448	405	463	673	255	438	427	506	378	480	556	218	423	329
NE_L_I (Mcal/day)	Mean	3.37	3.23	3.65	3.96	2.43	2.97	3.46	3.34	3.13	2.91	3.93	1.56	2.27	2.28
	Min	2.24	2.97	2.89	3.29	2.25	1.66	2.93	2.84	2.59	2.53	3.38	1.11	1.98	2.05
	Max	4.42	3.90	4.56	5.13	2.61	4.10	4.20	3.86	3.82	3.27	4.68	2.27	2.48	2.50
PDIN balance (g/day) ^3^	Mean	63.1	27.7	33.1	130.9	60.7	48.7	70.3	122.1	89.2	117.9	140.7	−31.2	112.5	81.4
	Min	36.7	−9.5	16.6	104.0	40.1	10.2	31.3	76.6	52.6	99.1	124.5	−25.5	75.8	−19.1
	Max	95.6	48.6	48.9	178.0	92.4	92.6	103.7	104.1	136.3	152.7	164.8	102.6	125.5	115.9
PDIN balance (%) ^4^	Mean	33.7	18.3	18.9	80.5	61.0	34.8	60.4	103.5	64.4	91.1	84.1	48.3	116.0	79.1
	Min	26.6	−4.6	11.7	72.6	29.6	10.6	20.5	57.8	37.4	60.9	63.0	−22.4	57.0	14.8
	Max	43.3	27.1	23.5	87.7	107.6	58.1	111.9	160.7	107.3	121.9	122.4	163.3	163.3	138.7
Milk (g/day)	Mean	1678	1393	1541	1445	728	934	990	899	1181	1075	1584	498	732	746
	Min	880	939	1084	1072	393	643	343	503	813	864	1052	249	386	422
	Max	2235	2133	1897	1782	1098	1218	1372	1242	1498	1450	2405	928	1126	986
Fat (%)	Mean	4.0	4.3	4.5	6.0	4.2	4.9	4.7	6.0	4.3	6.4	5.6	4.9	4.8	6.0
	Min	3.3	3.7	3.8	4.0	3.9	4.1	3.7	4.9	3.1	5.5	5.0	3.6	4.2	3.8
	Max	5.1	5.3	5.4	7.7	4.8	5.4	5.6	7.0	6.1	7.9	7.1	6.6	5.1	8.0
Protein (%)	Mean	5.1	5.4	5.1	4.9	5.4	5.7	5.3	5.2	5.3	5.0	5.0	5.8	4.8	5.4
	Min	4.7	4.7	4.8	4.1	4.8	5.1	4.5	4.4	4.2	4.5	4.7	3.9	4.6	4.8
	Max	6.0	6.0	5.5	6.0	6.0	6.2	6.3	6.9	7.1	5.8	5.9	6.9	5.4	6.6
Lactose (%)	Mean	5.1	5.0	4.9	4.8	4.4	4.8	4.5	4.6	4.4	4.6	4.7	3.5	4.3	4.3
	Min	4.8	4.8	4.8	4.7	4.2	4.5	4.0	4.2	4.0	4.5	4.2	2.8	3.5	3.8
	Max	5.3	5.2	5.1	5.0	4.8	5.1	4.8	5.1	4.9	4.9	5.1	4.4	5.0	4.7
MUC (mg/dL)	Mean	33.5	36.2	34.7	48.4	38.4	46.1	38.8	44.9	33.3	57.2	43.5	37.7	72.8	57.9
	Min	27.6	30.2	30.4	37.8	30.8	39.4	31.4	35.7	23.0	43.7	39.8	28.5	57.1	52.2
	Max	42.6	46.8	40.8	55.5	43.7	50.7	48.5	51.0	44.8	67.5	51.3	44.1	81.5	67.8

^1^ Diets (based on main ingredients): CF = corn flakes; BM = barley meal; BP = beet pulp; CC = corn cobs; CM = corn meal; AA = dehydrated alfalfa; SH = soybean hulls.; ^2^ DMI = dry matter intake; CPI = crude protein intake; NE_L_I = net energy for lactation intake; PDIN = protein digestible in the intestine when energy is not limiting rumen microbial growth; MUC = milk urea concentration.; ^3^ PDIN balance (g/day) = PDIN allowance (g/day) − PDIN requirement (g/day).; ^4^ PDIN balance (%) = (PDIN allowance (g/day) − PDIN requirement (g/day))/PDIN requirement (g/day) × 100.

**Table 4 animals-09-01118-t004:** Prediction of milk urea concentration (MUC, Y, mg/dL) according to linear fixed effects regression models based on diet composition or nutrient intake as independent regression variables using pooled data from experiments 1 and 2 conducted on mid- and late-lactation dairy sheep.

X ^2^	Intercept			Slope			Whole Equation ^1^			
	B_0_	s.e.	*p* <	C_0_	s.e.	*p* <	rmse ^5^	CV ^6^	R ^2^	*p* <
CP (% DM)	−68.4	61.6	0.29	6.1	3.3	0.09	10.46	23.5	0.15	0.09
A (% DM)	4.16	14.5	0.77	11.1	3.9	0.01	9.19	20.6	0.35	0.015
B_1_ (% DM)	19.7	7.1	0.02	17.1	4.7	0.003	8.14	18.3	0.49	0.003
A+B_1_ (% DM)	1.3	11.1	0.91	8.5	2.1	0.002	7.79	17.5	0.53	0.001
NDF (% DM)	19.3	10.7	0.09	0.6	0.3	0.03	9.70	21.8	0.27	0.03
ADF (% DM)	23.9	6.8	0.004	0.9	0.3	0.007	8.68	14.5	0.42	0.007
ADL (% DM)	19.6	6.1	0.007	7.4	1.7	0.001	0.61	16.7	0.57	0.001
NFC (% DM)	68.9	8.2	0.001	−0.7	0.2	0.009	8.85	19.9	0.39	0.009
starch (% DM)	58.9	5.3	0.001	−0.6	0.2	0.01	8.88	19.9	0.39	0.01
NFCI (g/day)	76.1	11.2	0.001	−0.05	0.02	0.01	9.09	20.4	0.36	0.01
starchI (g/day)	64.5	5.9	0.001	−0.05	0.01	0.003	8.19	19.4	0.48	0.003
NE_L_ (Mcal/kg DM)	127.6	9.6	0.001	−51.2	5.8	0.001	4.36	9.8	0.85	0.001
CP/NE_L_ (g/Mcal)	−14.9	7.4	0.07	0.5	0.06	0.001	4.63	10.4	0.83	0.001
PDIN balance (g/day) ^3^	32.2	6.4	0.001	0.15	0.07	0.056	10.10	22.6	0.27	0.056
PDIN balance (%) ^4^	26.9	5.1	0.001	0.28	0.07	0.003	7.99	17.96	0.54	0.003

^1^ Regressions based on experimental treatment means.; ^2^ X = Independent regression variable; CP = crude protein; B_0_ is the intercept, C_0_ is the regression coefficient, A = non-protein nitrogen (NPN); B_1_ = buffer soluble true protein; NDF = neutral detergent fiber; ADF = acid detergent fiber; ADL = acid detergent lignin; NFC = non-fiber carbohydrates; NFCI = NFC intake; starchI = starch intake; NE_L_ = net energy for lactation; PDIN = protein digestible in the intestine, when energy is not limiting rumen microbial growth.; ^3^ PDIN balance (g/day) = PDIN intake (g/day) − PDIN requirement (g/day); ^4^ PDIN balance (%) = (PDIN intake (g/day) − PDIN requirement (g/day))/PDIN requirement (g/day) × 100; ^5^ rmse = root of the mean squared error; ^6^ CV = coefficient of variation.

**Table 5 animals-09-01118-t005:** Relationship between milk urea concentration (mg/dL; dependent variable) and other variables according to either linear regression or mixed regression models inclusive of the “study effect” (Exp) using pooled data of eight experiments on stall-fed dairy sheep.

X ^2^	Intercept			Slope			Whole Equation ^1^			
	A_0_	s.e.	*p* <	B_0_	s.e.	*p* <	rmse ^3^	CV ^4^	R ^2^	*p* <
CP (% DM)	−25.3	7.5	0.01	3.9	0.4	0.001	7.35	15.97	0.68	0.001
CP Exp (% DM)	−20.7	8.6	0.05	3.7	0.5	0.001	4.89	10.62	0.82	0.001
NDF (% DM)	43.1	11.7	0.001	0.07	0.3	0.79	13.11	28.5	0.001	0.79
NDF Exp (% DM)	32.3	9.2	0.009	0.4	0.2	0.07	5.60	12.2	0.64	0.001
NFC (% DM)	68.2	7.6	0.001	−0.6	0.2	0.006	9.98	21.50	0.22	0.006
NFC Exp (% DM)	69.75	6.9	0.001	−0.6	0.2	0.001	4.89	10.52	0.56	0.001
NE_L_ (Mcal/kg DM)	145.5	16.7	0.001	−62.4	10.4	0.001	9.64	21.00	0.46	0.001
NE_L_ Exp (Mcal/kg DM)	124.1	16.2	0.001	−48.5	10.1	0.001	5.46	11.84	0.73	0.001
CP/NE_L_ (g/Mcal)	−15.1	3.6	0.001	0.5	0.03	0.001	4.56	9.90	0.88	0.001
CP/NE_L_ Exp (g/Mcal)	−13.7	4.2	0.014	0.5	0.03	0.001	3.30	7.17	0.93	0.001
CPI (g/day)	23.9	6.5	0.001	0.05	0.01	0.001	11.54	25.06	0.23	0.001
CPI Exp (g/day)	17.8	6.9	0.03	0.07	0.01	0.001	5.24	11.38	0.76	0.001

^1^ A reduced model without covariance components was implemented. Full model was not estimable.; ^2^ X is the independent regression variable, A_0_ is the overall intercept, and B_0_ the overall slope; n = 44 treatment means; ^3^ rmse = root of the mean squared error; ^4^ CV = coefficient of variation.

**Table 6 animals-09-01118-t006:** Milk urea concentrations (MUC) as a function of dietary energy (NE_L_) and protein (CP) concentrations based on the following equation: MUC (mg/100 mL) = −13.7 + 0.5 CP/NE_L_ (g/Mcal). The grey area (numbers in italics) is at risk of decreases in sheep fertility and impaired sheep health in the long term.

Dietary NE_L_ (Mcal/kg DM)	Dietary CP (g/kg DM)								
	120	130	140	150	160	170	180	190	200
1.2	36	40	45	49	53	57	61	65	70
1.3	32	36	40	44	48	52	56	59	63
1.4	30	33	36	40	43	47	51	54	58
1.5	27	30	33	36	40	43	46	50	53
1.6	25	27	30	33	36	39	43	46	49
1.7	22	25	27	30	33	36	39	42	45

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
