# Peer review of "Milk Urea Concentration in Dairy Sheep: Accounting for Dietary Energy Concentration"

_animals, 2019, doi:10.3390/ani9121118_

Round 1

Reviewer 1 Report

General comments

The topic of this paper (Revisited reference values for milk urea concentration in dairy sheep: accounting for dietary energy concentration) falls within the general scope of the journal. The aim of this study was to assess the effects on milk urea concentration (MUC) of diets differing for energy concentration and carbohydrate sources in lactating dairy ewes. In general, the manuscript is clear, with an exhaustive introduction, updated to the most recent works. The aims are clearly defined. Material and methods section is well described considering all elements and aspects necessary to comprehend the experimental work, and an appropriate statistical analysis was applied. The results, divided into two different parts, report the main values presented in the tables and the figures in a proper way.

Only a remark regarding the title of the manuscript is required: a simplify title (Milk urea concentration in dairy sheep accounting for dietary energy concentration) should be more appropriate in relation to the content of the manuscript.

Specific comments

ABSTRACT

Line 35: please, change as follows: …MUC = 127.6 - 51.2 × NEL …

MATERIAL and METHODS

Line 100: table 1 needs to be reformatted.

Line 107: please, change as follows: … and CC = corn cobs, in E1; and CM = corn meal;….

Line 108: please, change as follows: … and SH = soybean hulls, in E2.

Line 112 (table 2): please, for completeness of information, indicate (in the table footnotes) the minerals and vitamins composition of the integrator. Moreover, in this table, it should be advisable to report the concentration values of PDIE and PDIN of the different treatments.

Line 134: please, change as follows: … by weighing offered diets …

Line 136: please, indicate the frequency of milk yield recordings.

Line 137: please, regarding milk sampling, indicate when the milk samples were taken (morning, afternoon, both). If milk samples were taken only at the morning milking, the milk fat concentration was probably less than the fat concentration of the milk produced in the entire day (24h) because of the higher time interval between afternoon and morning milking.

Line 144: please, indicate if NDF, ADF, and ADL, were expressed as ash free values. Please, indicate if the NDF procedure included the use of sodium sulfite.

DISCUSSION

Line 353-355 (table 6): the statement reported in the title of this table regarding the numbers of MUC in the grey area is not clearly supported in the discussion; the reference value of MUC (560 mg/L) of the reference [35] cited in the manuscript could be explicitly reported in the text sustaining the fact that the “grey” values could decrease sheep fertility.

Lines 360-361: this sentence is not clear. What do the authors mean with “probably causing a waste of nitrogen”? The authors could give more comments at this regard, also in relation to the sheep requirements of PDI and the MUC.

REFERENCES

Line 448: please, change as follows: Ubertalle, A.; Fortina, R.; …

Author Response

Dear reviewer,

we thank you for your valuable comments, they helped us to improve the paper.

The Authors

General comments

The topic of this paper (Revisited reference values for milk urea concentration in dairy sheep: accounting for dietary energy concentration) falls within the general scope of the journal. The aim of this study was to assess the effects on milk urea concentration (MUC) of diets differing for energy concentration and carbohydrate sources in lactating dairy ewes. In general, the manuscript is clear, with an exhaustive introduction, updated to the most recent works. The aims are clearly defined. Material and methods section is well described considering all elements and aspects necessary to comprehend the experimental work, and an appropriate statistical analysis was applied. The results, divided into two different parts, report the main values presented in the tables and the figures in a proper way.

Only a remark regarding the title of the manuscript is required: a simplify title (Milk urea concentration in dairy sheep accounting for dietary energy concentration) should be more appropriate in relation to the content of the manuscript.

Done: we changed the title as suggested.

Specific comments

ABSTRACT

Line 35: please, change as follows: …MUC = 127.6 - 51.2 × NEL …

Done

MATERIAL and METHODS

Line 100: table 1 needs to be reformatted.

Done: we reformatted Table 1 as requested.

Line 107: please, change as follows: … and CC = corn cobs, in E1; and CM = corn meal;….

Done

Line 108: please, change as follows: … and SH = soybean hulls, in E2.

Done

Line 112 (table 2): please, for completeness of information, indicate (in the table footnotes) the minerals and vitamins composition of the integrator. Moreover, in this table, it should be advisable to report the concentration values of PDIE and PDIN of the different treatments.

Done: mineral and values of PDIN and PDIE has been added in Table 2.

In the same table a feed was added, forgotten during the preparation of the first table, and the dietary PDIE and PDE values of the treatments were modified accordingly

Line 134: please, change as follows: … by weighing offered diets …

Done: I changed the word “concentrates” with “diets”.

Line 136: please, indicate the frequency of milk yield recordings.

Done: the frequency of milk yield recordings has been added.

Line 137: please, regarding milk sampling, indicate when the milk samples were taken (morning, afternoon, both). If milk samples were taken only at the morning milking, the milk fat concentration was probably less than the fat concentration of the milk produced in the entire day (24h) because of the higher time interval between afternoon and morning milking.

Done: this information has been added in the text.

Line 144: please, indicate if NDF, ADF, and ADL, were expressed as ash free values. Please, indicate if the NDF procedure included the use of sodium sulfite.

Done: this information has been added in the text.

DISCUSSION

Line 353-355 (table 6): the statement reported in the title of this table regarding the numbers of MUC in the grey area is not clearly supported in the discussion; the reference value of MUC (560 mg/L) of the reference [35] cited in the manuscript could be explicitly reported in the text sustaining the fact that the “grey” values could decrease sheep fertility.

Done: A  sentence has been added in the text

Lines 360-361: this sentence is not clear. What do the authors mean with “probably causing a waste of nitrogen”? The authors could give more comments at this regard, also in relation to the sheep requirements of PDI and the MUC.

Done: The concept has been clarified.

 REFERENCES

Line 448: please, change as follows: Ubertalle, A.; Fortina, R.; …

Done

Reviewer 2 Report

Minor errors and text editing:

Line 35: Missing a +. MUC=127.6+51.2XNEL

Line 100: Correct Table 1 format. There are numbers and parentheses in rows that do not belong

Line 112: Correct Table 2. Put the units of the Ingredients in the same way as those of Chemical composition.

Line 197: Write starch with a lowercase letter, to unify with the rest of the text was it always appears in lowercase.

Line 253: Although the abbreviation of the word versus may be v. or vs. I think the last one is more common.

Methodological errors:

The paper has two main objectives, the first is well described but the second can be improved in the part related to the meta-analysis methodology.

How was the papers search done? (Search engines, words used in the search, years investigated, etc.)

Number of papers found.

Inclusión criteria and number of papers discarded for each inclusión criteria not match.

Author Response

Dear Reviewer,

we thank you for your valuable and constructive comments.

The Authors

Line 35: Missing a +. MUC=127.6+51.2XNEL

 Done

Line 100: Correct Table 1 format. There are numbers and parentheses in rows that do not belong

Done: we reformatted Table 1 as requested.

Line 112: Correct Table 2. Put the units of the Ingredients in the same way as those of Chemical composition.

Done

Line 197: Write starch with a lowercase letter, to unify with the rest of the text was it always appears in lowercase.

Done

Line 253: Although the abbreviation of the word versus may be v. or vs. I think the last one is more common.

Done

Methodological errors:

The paper has two main objectives, the first is well described but the second can be improved in the part related to the meta-analysis methodology.

How was the papers search done? (Search engines, words used in the search, years investigated, etc.)

Number of papers found.

Inclusion criteria and number of papers discarded for each inclusion criteria not match.

Done

Reviewer 3 Report

The study is very interesting, correctly designed and presented, extensive and practice orientated.

I have no concern regarding it; just some comments or suggestions, which authors would like to add or modify.

I suggest to review these two references of the present Journal Animals, where some values of high producing ewes are reported, although not describing or linking it to the nutritional level, interesting because there are scarce information on this:

Pesantez-Pacheco et al. (2019) Maternal Metabolic Demands Caused by Pregnancy and Lactation: Association with Productivity and Offspring Phenotype in High-Yielding Dairy Ewes. Animals (Basel). 2019 May 30;9(6). pii: E295. doi: 10.3390/ani9060295.

AND

Pesantez-Pacheco et al. (2019). Influence of Maternal Factors (Weight, Body Condition, Parity, and Pregnancy Rank) on Plasma Metabolites of Dairy Ewes and Their Lambs. Animals (Basel). 2019 Mar 28;9(4). pii: E122. doi: 10.3390/ani9040122.

Author Response

Dear Reviewer,

we thank you for your valuable and constructive comments.

The Authors

Line 23: First time authors use this abbreviation. Please describe

Done

Line 23: "MUC of diets" does not exist, It is MUC of the ewes which received diets....

Please reword this in a more clear way

Done: The sentence has been rephrased.

Line 84: Describe please the age/lactations of these ewes, and the distribution of lactation per ewe of each experimental group

Line 100: Tables should be autoexplicative! All descriptions of the abreaviations (written in lines 94-99) should be written under the table.

Please take care of the presentation of the table: the nimbers does not fit in the places, and makes the table difficult to follow

Done: we reformatted Table 1 as requested.

Describe  age/lactations of the ewes included, and the distribution of lactation per ewe of each experimental group

Done

Line 190: I would put these titles not in the center of the raw, but in the first line of each row, in order to make esier to see the first line  (values of the Mean) of each item

Done

Line 354: Format of these figures is different of that used in the rest of tables and figures

Corrected

Line 396: two additional references suggested with urea levels of lactating ewes described.

Pesantez-Pacheco et al. (2019) Maternal Metabolic Demands Caused by Pregnancy and Lactation: Association with Productivity and Offspring Phenotype in High-Yielding Dairy Ewes. Animals (Basel). 2019 May 30;9(6). pii: E295. doi: 10.3390/ani9060295.

Pesantez-Pacheco et al. (2019). Influence of Maternal Factors (Weight, Body Condition, Parity, and Pregnancy Rank) on Plasma Metabolites of Dairy Ewes and Their Lambs. Animals (Basel). 2019 Mar 28;9(4). pii: E122. doi: 10.3390/ani9040122.

Thank you for the suggestions. However, the papers above, although interesting, refer to plasma and not milk urea and, what is more important, do not report data on intake of nutrients or their dietary proportions. Moreover, their inclusion in the discussion chapter of this paper it is not easy because the factors addressed in the Pesantez-Pacheco et al., papers such as parity, pregnancy rank go beyond the scope of our study. Other factors that are ascribed to milk yield, BW and BCS in those papers, could be possibly explained by higher intakes or dietary proportion of protein, hence we are unable to compare our with their results.